# Comparative Analysis of Corneal Parameters Performed with GalileiG6 and OCT Casia 2

**DOI:** 10.3390/diagnostics13020267

**Published:** 2023-01-11

**Authors:** Robert Mazur, Adam Wylęgała, Edward Wylęgała, Dariusz Dobrowolski

**Affiliations:** 1Chair and Clinical Department of Ophthalmology, Division of Medical Science in Zabrze, Medical University of Silesia, 40-055 Katowice, Poland; 2Pathophysiology Department, School of Medicine, Medical University of Silesia, 40-055 Katowice, Poland

**Keywords:** keratometry, central corneal thickness, SS-OCT CASIA 2, Dual Scheimpflug Analyzer GalileiG6, astigmatism, thinnest corneal thickness, pachymetry

## Abstract

Backgrounds: To compare keratometry (Ks and Kf), astigmatism (Ast.), and the astigmatism axes (Ax.) of the posterior surface of the cornea; the total, central cornea thickness (CCT); and the thinnest corneal thickness (TCT) measured using two different measurement methods. Methods: Patients qualified for cataract surgery at the Chair and Clinical Department of Ophthalmology, Division of Medical Science in Zabrze, Medical University of Silesia, Katowice, Poland, were included in the study and monitored with the following two devices: OCT-CASIA2 and Dual Scheimpflug Analyzer GalileiG6. Our work was a randomized, prospective study in which compliance with the agreement of measurements between the devices was evaluated using the Bland–Altman method. Results: A total of 110 patients (62 females and 48 males) were examined. Overall, 100 eyes of patients that qualified for cataract surgery were enrolled in the study. No statistically significant difference was observed for Total-Ks and Total-Kf. A significant difference was observable for the following parameters: total Ks-ax, total Kf-ax, the total power of astigmatism, and in all parameters of the part of the cornea and corneal thickness (CCT and TCT). Conclusions: The measurements obtained using Casia2 and the Dual Scheimpflug Analyzer GalileiG6 were significantly different and not interchangeable except for total Ks and Kf.

## 1. Introduction

Precise biometric measurements underpinning the calculation of the intraocular lens (IOL) power are the most crucial aspect of cataract surgery. During the examination of the patient before cataract surgery, the depth of the anterior chamber and keratometry is measured each time. Measurements of these parameters are used to determine the power of the artificial intraocular lens using automatic algorithms in measuring devices. Our previous work focused on the broader aspect of calculating an artificial intraocular lens during qualification for cataract surgery. We compared total keratometry, anterior chamber depth, central pachymetry, and eyeball axial length using most of the available measurement methods [1]. The current study focuses mainly on the anterior segment of the eye, i.e., total keratometry (total steep keratometry in diopters—Ks (D), total flat keratometry in diopters—Kf (D)), posterior keratometry (posterior steep keratometry in diopters—Ks (D), posterior flat keratometry in diopters—Kf (D)), central corneal thickness, thinnest corneal thickness, total astigmatism, posterior corneal astigmatism, and astigmatism axis. In the modern treatment of cataracts, surgeons need to customize the refractive prediction of different patients to meet their visual expectations after cataract surgery. Therefore, an accurate intraocular lens (IOL) calculation formula is essential. Olsen T pointed out that keratometry (K) measurement errors accounted for 22% of the total prediction error and the ACD measurement error accounted for 42%. Therefore, accurate preoperative ocular biometry is highly important for patients with cataracts to obtain a good refractive status [2,3,4]. Various types of IOL, including monofocal, multifocal, toric, and enhanced depth-of-focus, have been developed to provide the expected level of vision. The accuracy of IOL calculations, achieved with a suitable ocular biometer, is essential for the success of the procedure [5,6]. The accuracy of measurements of corneal parameters is even more critical in terms of the use of toric and multifocal lenses. In the case of an error in measuring the axis and the power of astigmatism, we can obtain a deepening of the cylindrical defect, which significantly reduces the comfort of life after cataract surgery. In our project, we wanted to compare the following two measurement systems for examining the anterior segment of the eye: the GalileiG6 Dual Scheimpflug Analyzer (Ziemer, Port, Switzerland) and CASIA2 Swept Source OCT (AS-OCT) (Tomey Corporation, Nagoya, Japan). Both devices have different measuring methods. Our study aims to assess whether all or some of the measurable parameters can be used interchangeably during cataract surgery.

## 2. Materials and Methods

Patients qualified for cataract surgery at the Chair and Clinical Department of Ophthalmology, Division of Medical Science in Zabrze, Medical University of Silesia, Katowice, were consecutively included. This prospective study complied with the requirements of the Declaration of Helsinki and was approved by the Ethics Committee of the Division of Medical Science in Zabrze, Medical University of Silesia (Commission approval number KNW/0022/KB1/128/18/19, date of approval 08.01.2019). All participants provided their informed consent and received a leaflet explaining the nature of the study. Patients referred for cataract surgery as the sole indication for surgery were eligible for the study. Minors and pregnant women did not participate in the study. Prior to measurements, the patients underwent a complete ophthalmic examination. Criteria for exclusion from the study were established, which may affect errors in the obtained results, such as corneal scars, corneal dyscrasias, keratopathies, previous refractive surgery, condition after glaucoma surgery, condition after vitreoretinal surgery, severe dry eye syndrome, gerontoxone, corneal pterygia and the condition after injuries to organs of vision. A total of 110 eyes of patients who qualified for cataract surgery were examined consecutively on the day before surgery. A total of 62 females and 48 males were examined. The mean age was 66 years, with a range of 45 to 85 years. Cylindrical power was as follows: mean 1.06; min 0.05; max 2.91; and Std.Dev. 0.65. The axis was as follows: Mean 85.09; min 3.00; max 169.00; and Std.Dev. 22.22. In total, 51 right and 49 left eyes were included in the study. Only one eye of each patient was included in the study.

To reduce the influence of external factors on the measurements, the devices were placed in the same darkened room. Measurements were taken the day before the scheduled surgical treatment. This allowed for the exclusion of external factors that could affect the measurement error, such as the level of illumination and air humidity affecting the condition of the tear film. Measurements on both devices were performed consecutively at the same time of day at around noon. One operator performed all tests. The researcher is an ophthalmologist with several years of experience; before the examination, he performed measurements with both devices. At the time of the examination, only the patient and the researcher were in the room. Inclusion in the study was conditional on correct measurements with all devices. Only measurements with high measurement quality indicators dictated by the devices were included in the study. Ten patients were excluded from the study due to corneal changes that could affect the measurement quality, three due to post-inflammatory changes of the cornea, one due to keratoconus, one due to previous refractive surgery, and five due to gerontoxon.

Study devices characteristics:

1. The GalileiG6 Dual Scheimpflug Analyzer (Ziemer, Port, Switzerland) combines topography based on 20 Placido rings with a dual rotating Scheimpflug camera. Simulated keratometry is calculated from the annular (half-cord) zone from 0.5 to 2.0 mm and diopters at a refractive index of 1.3375 were observed. The posterior mean K was calculated using a refractive index of 1.376 for the cornea and 1.336 for aqueous humor, which is calculated over a 4 mm diameter area [7]. Axial biometry is performed using an 880 nm light based on low coherence interferometry [1,6,8]. Placido imaging provides high-accuracy curvature data, while Scheimpflug imaging is optimal for precise elevation data. Galilei captures slit images from the opposite sides of the illuminated slit and averages the elevation data obtained from the corresponding opposite slit images. This Dual Scheimpflug Imaging technique improves the detection of the posterior corneal surface and provides outstanding accuracy in pachymetry across the entire cornea, even when the camera is decentered due to eye movements. Placido imaging provides high-accuracy curvature data, while Scheimpflug imaging is optimal for precise elevation data. Galilei captures slit images from opposite sides of the illuminated slit and averages the elevation data obtained from opposite slit images. This Dual Scheimpflug Imaging technique improves the detection of the posterior corneal surface and provides outstanding accuracy in pachymetry across the entire cornea, even when the camera is decentered due to eye movements [5,9].

Although the resolution of Scheimpflug images is high enough to deliver accurate profile data, it is insufficient to calculate central corneal power (curvature data) with acceptable accuracy. Galilei overcomes this limitation by merging Placido and Scheimpflug data, acquired simultaneously by the two techniques, into a comprehensive single data set. This is essential for obtaining the highest accuracy for both elevation and curvature data across the entire cornea. The measurement is made with 20 Placido rings at a speed of 60 scans per second, with an accuracy of up to 100 000 measurement points. The Scheimpflug camera resolution is 2 × 1280 × 960 [10] (Figure 1B).

2. The CASIA2 Swept Source OCT (AS-OCT) (Tomey Corporation, Nagoya, Japan) uses a 1310 nm swept laser wavelength, which is longer than in SD-OCT devices, providing higher penetration but lower resolution; the fast detector 50,000 Ascans/s, uses a CMOS camera. Axial resolution (tissue): 10 μm; resolution after transverse: 30 μm; scan range: 12 mm; scan depth: 13 mm; shaft-image acquisition bone: 0.3–2.4 s [11,12]. Corneal power is calculated using a refractive index of 1.3375. In addition, keratometry values were calculated for a diameter of 3.2 mm [1,13]. The anterior and posterior cornea curvature is measured by performing a radial scan of 16 images, with a scan resolution of 800 A-scans per line sampling, the scan speed of 0.3 s, scan range of 16 mm, and depth of 11 mm [14] (Figure 1A).

Statistical analysis: Data were collated in a LibreOffice 7.0 spreadsheet. The comparison of the results of the two repeated measurements was made using the Wilcoxon test for the pairing pairs. The compliance of the measurements was assessed using the ICC intraclass correlation coefficient and the Bland–Altman method [15]. The limit of agreement (LoA) was calculated as ±1.96 × SD of the difference. The correlation is perfect when its value is close to 1. We considered an ICC of close to 1 as excellent and we considered everything above 0.991 to be high. A significance level of 0.05 was adopted in the analysis. Thus, all *p*-values below 0.05 were interpreted as showing significant relationships. The analysis was performed in the R software, version 4.0.4. (R Core Team (2021). R: A language and environment for statistical computing. R Foundation for Statistical Computing, Vienna, Austria).

## 3. Results

A total of 110 eyes of patients that qualified for cataract surgery were examined. Due to the detection of coexisting diseases such as post-inflammatory scarring of the cornea, keratoconus, and gerontoxon status after refractive surgery, 100 eyes were ultimately included from the 110 patients. A comparison of individual devices was made. The following results were obtained. The sample size was calculated using a power goal of 0.9 and alpha of 0.05. The standardized effect was 0.4, and the required sample size was 68.

### 3.1. Total Corneal Power

In the comparison, the total Ks (D) and Kf (D) of the CASIA2 Swept Source OCT device and GalileiG6 Dual Scheimpflug Analyzer were studied, and no significant differences between devices were confirmed (*p* > 0.05). The mean difference between devices is small. A high ICC was demonstrated (Table 1 and Figure 2).

In the OCT Casia 2 and GalileiG6 Dual Scheimpflug Analyzer comparison, there were significant differences between devices for the total Ks axis (°). The difference is insignificant (*p* > 0.05), and the ICC is low (Table 1 and Figure 2).

In terms of the total Kf axis (°) in the comparison, there were significant differences between devices. The difference is insignificant (*p* > 0.05) and ICC is low (Table 1 and Figure 2).

In terms of total astigmatism As.(D) in the Casia 2 and GalileiG6 Dual Scheimpflug Analyzer comparison, there were significant differences between devices. The difference is significant (*p* >0.05) and ICC is low (Table 1 and Figure 2).

### 3.2. Posterior Corneal Power

When comparing the posterior corneal parameters Ks (D) and Kf (D) for the CASIA2 Swept Source OCT device and GalileiG6 Dual Scheimpflug Analyzer, a significant difference was observed (*p* < 0.05) and the ICC was low (Table 2 and Figure 3).

In the comparison of the posterior Kf axis (°) and Ks axis (°) for the OCT Casia 2 device and GalileiG6 Dual Scheimpflug Analyzer, there were significant differences between devices. The difference is significant (*p* < 0.05), and the ICC is low (Table 2 and Figure 3). Likewise, significant differences were also found in the total magnitude of posterior astigmatism As.(D) (*p* < 0.05) and the ICC was low (Table 2 and Figure 3).

### 3.3. Corneal Thickness

Comparing the central pachymetry of CCT (um) between the OCT Casia 2 device and GalileiG6 Dual Scheimpflug Analyzer, the following results were obtained: The difference is significant (*p* < 0.05), and ICC is moderately high (Table 3 and Figure 4).

Due to the finding of discrepancies in the measurements of the astigmatism axes Ks and Kf, we performed repeatability measurements of the astigmatism axes for GalileiG6 Dual Scheimpflug Analyzer and CASIA2 Swept Source OCT. The repeatability of the total astigmatism axis Ks for the GalileiG6 Dual Scheimpflug Analyzer was: Mean 86.09, Std.Dev. 86.09, ICC 0.92. The repeatability of the total astigmatism axis Ks for CASIA2 Swept Source OCT was: Mean 86.27, Std.Dev. 57.09, ICC 0.92. The repeatability of the Ks posterior astigmatism axis for the GalileiG6 Dual Scheimpflug Analyzer was: Mean 82.63, Std.Dev. 32.80, ICC 0.92. The repeatability of the posterior astigmatism axis Ks for CASIA2 Swept Source OCT was: Mean 87.54, Std.Dev. 18.46, ICC 0.87.

## 4. Discussion

In this study, a Bland–Altman analysis was performed to compare two ocular measuring devices. The study is instrumental in helping users to determine which platforms can be used interchangeably. In addition, the test is helpful for cataract surgery. We wanted to determine possible differences in the values of the anterior segment of the eye and prove whether both methods can be used interchangeably in cataract surgery.

Our previous study proved the possibility of the interchangeable use of GalileiG6 and IOL Master 500 during qualification for cataract surgery. The differences between AL, ACD, K1, and K2 measurements are insignificant; therefore, they should not affect the power of the implanted lens during cataract surgery. We also proved the convergence of AL. Measurements were performed with the above devices vs. USG Quantel. However, there are wide differences in the ACD, and keratometry measurements between GalileiG6 and IOL Master 500 vs. USG Quantel and OCT Casia 2. This suggests that these devices should not be used interchangeably for biometric measurements and IOL power calculations [1].

In the present study, we focused on a more detailed analysis of the parameters of the anterior segment of the eye during qualification for cataract surgery. In addition, focusing on the parameters of the cornea, we wanted to identify possible deviations that may result in postoperative refractive errors when using devices interchangeably during cataract surgery qualification.

In the comparison of total Ks (D) and Kf (D) for CASIA2 Swept Source OCT vs. GalileiG6 Dual Scheimpflug Analyzer, no significant differences between devices were confirmed. The above results prove that these two devices provide similar results to be used interchangeably. For the total and posterior Ks axis (°) and Kf axis (°), total and posterior astigmatism As.(D), posterior corneal parameters Ks (D), and Kf (D) in the CCT OCT Casia 2 and GalileiG6 Dual Scheimpflug Analyzer comparison, there were significant differences between devices. Therefore, Galilee and Casia 2 are not compatible in terms of these parameters. Additionally, we noticed a tendency to overestimate the results of total and posterior astigmatism As. (D) by GalileiG6. We also found a tendency for the GalileiG6 to overestimate CCT, suggesting that the parameters should not be used interchangeably between devices.

Casia 2 is a Swept Source anterior segment OCT. Corneal power is calculated using a 1.3375 refractive index. Further, keratometry values are calculated on a 3.2 mm diameter. Contrary Galileli combines 20 Placido rings with a dual rotating Scheimpflug camera. Simulated keratometry (SimK) is calculated from the 0.5 to 2.0 mm annular (semichord) zone and is represented as diopters using a refractive index of 1.3375 [2,13].

As we found in previous studies, the GalileiG6 has a tendency to overestimate keratometry at higher levels of astigmatism. Measuring the anterior corneal surface is easier than measuring the posterior. In order to measure the latter, sophisticated mathematic algorithms have to be implemented, which is why there is a significant difference between the recordings of the devices. Secondly, due to the strong reflex at the air/cornea interface, it is difficult to correctly identify edges. Thirdly, posterior surface evaluation is hindered by the errors of the front surface. Moreover, the size of the posterior measurement is different for the two devices. Casia 2 measures a 3.2 mm radius while GalileiG6 measures a 4 mm radius. The refractive indexes for the posterior or surface can vary for different devices. Anterior surface keratometry can be measured in simulated keratometry when values are calculated from the annular (semichord) or in true keratometry where values are measured within the circle. There is no posterior simulated keratometry [13].

We noticed similar results in previous studies. Yune Zhao et. al. compared the CASIA ssOCT (SS-1000; Tomey, Nagoya, Japan) with the Scheimpflug camera imaging device, the Pentacam (Oculus Optikgeräte GmbH, Wetzlar, Germany), in a similar analysis. The study’s conclusions were as follows: The precision of CT measurements made by SS-OCT was higher, while the reliability of keratometry measurements using the Scheimpflug system was higher in children. Apart from the measured values in the central corneal region, the thickness and keratometry readings should not be considered interchangeable between the two systems [16]. We found significant differences in the obtained parameters in our work, except for the total Ks and Kf. This suggests that only these parameters can be used interchangeably. Eszter Szalai et al. compared rotating Scheimpflug imaging and ss-OCT, and found significant differences in the keratometry, pachymetry, and ACD results between AS-OCT and Scheimpflug imaging. However, the repeatability of the measurements was comparable. The obtained results of the comparison of the measurement methods are similar to ours, but it should be noted that the researchers compared a group of people with keratoconus, which may impact measurement errors in relation to the healthy corneas tested in this study [17]. We also found similarity between our results and those of Javier González-Pérez et al., who concluded that while both devices appeared to be reliable, the differences between the devices regarding the measured tomographic parameters mean that the Pentacam HR and Casia 2 measurements are not interchangeable with either healthy corneas or KC [18]. In our study, we also confirmed previous research results suggesting that OCT tools tend to underestimate CCT values compared to Scheimpflug-based devices in normal corneas [19,20,21]. This is the result of the method of measuring the compared devices. Scheimpflug devices measure the thickness of the cornea between the tear film surface and air and the posterior surface of the cornea based on elevation points captured with a rotating camera [22]. Optical coherence tomography identifies the anterior and posterior surfaces of the cornea and converts the distance between them into the thickness of the cornea.

However, we found discrepancies in the results of the research by Yong Woo Lee et al. In their analysis, the researchers compared corneal topography measurements obtained using Scheimpflug-Placido dual rotary systems (Galilei G2). The OCT was of a variable source (Casia SS-1000) and the same was the case for the Placido-scanning-slit (Orbscan IIz). It was concluded that the anterior keratometry obtained using three devices showed high degrees of agreement. Posterior keratometry and eccentricity showed more remarkable agreement between the dual rotating Scheimpflug-Placido and swept-source OCT systems than with the Placido-scanning-slit system. Therefore, the dual rotating Scheimpflug-Placido and swept-source OCT systems were equivalent in detecting the shape of the cornea and could be considered interchangeable [23].

It should be emphasized that the measurements of the parameters of the anterior segment of the eye may differ between even very similar devices. It has been confirmed many times that measurement systems present such different results that they cannot be used interchangeably [24,25,26].

Therefore, it is essential to carry out research using more measuring devices in the future. These tests should focus not only on groups of devices using the same measurement method but also on different methods. An additional aspect is also checking these devices for the correctness of measurements in the case of diseases accompanying the anterior segment of the eye, such as corneal ectasia or post-inflammatory scars. Moreover, in the future, more patients will undergo cataract surgery who have previously undergone refractive surgery; therefore, it will become more critical to accurately measure the total power of the cornea [27,28]. Although the axial length has the greatest impact on refraction, given that the measurement error of 1 mm AL causes a deviation of 2.5 D in the calculation of the IOL in the eye with the average AL (23.5 mm) [29,30], it should be kept in mind that the measurement error of keratometry (K) accounted for 22% of the total prediction error [4]. This is important for the final clinical condition of the patient after cataract surgery and can significantly affect the elimination of refractive errors, which affects the quality of life.

Previous studies checking the repeatability and reproducibility of the devices we compared showed good reproducibility for OCT Casia2. Biswas et. all. showed in a group of healthy people that Casia, TMS-5, and Pentacam can be used interchangeably to measure corneal thickness and radius measurement in healthy eyes. Casia had the best agreement with ultrasound pachymeter CCT and exhibited the highest repeatability [31]. Additionally, OCT has been observed to have an advantage for corneal imaging in eyes with keratoconus. Chan et. all in their study showed significant differences in posterior corneal surface and corneal thickness measurements between swept-source OCT and combined Placido-Scheimpflug imaging in eyes with keratoconus. Swept-source OCT might be preferred over Placido-Scheimpflug imaging owing to the better repeatability of measurements [32]. Whereas Matar et. all. in a paper comparing ss-OCT and Pentacam devices, he observed that Both Casia 2 and Pentacam enable a reliable assessment of the corneal refractive power in KC after fs-INTACS implantation; however, the reproducibility was significantly better with Casia 2 only for the measurement of corneal thickness. The Pentacam showed significantly higher values for the mean anterior and posterior corneal refractive power and measured significantly thicker at the thinnest point of the cornea compared to Casia 2 [33].

These results suggest that ss-OCT may be more useful in measuring the anterior segment of the eye also before cataract surgery.

Limitations of the study: Our study group did not include measurements of patients with corneal disorders. Therefore, the results obtained in the above group may differ significantly compared to eyes with healthy corneas. This is related to the possible instability of the measurements, especially in the case of coral ectasia. In addition, the way that differences in the measurements of the examined parameters affect the clinical condition, especially the power of the artificial intraocular lens used during cataract surgery, should be investigated in the future.

## 5. Conclusions

Our study showed the possible interchangeable use of the total keratometry parameters of Ks (D) and Kf (D) when measured with the Dual Scheimpflug Analyzer GalileiG6 and SS-OCT CASIA 2. The differences between Ks (D) and Kf (D) are so insignificant that no effects should be observed in terms of clinical status when these parameters are used, including when calculating the IOL during cataract surgery.

Statistical differences were found in the case of the remaining parameters, including the axis and total values of astigmatism, the central and thinnest corneal thicknesses, and all parameters of the posterior part of the cornea we examined. This suggests that these devices are not to be used interchangeably, as differences may have an impact on the clinical status.

## Figures and Tables

**Figure 1 diagnostics-13-00267-f001:**
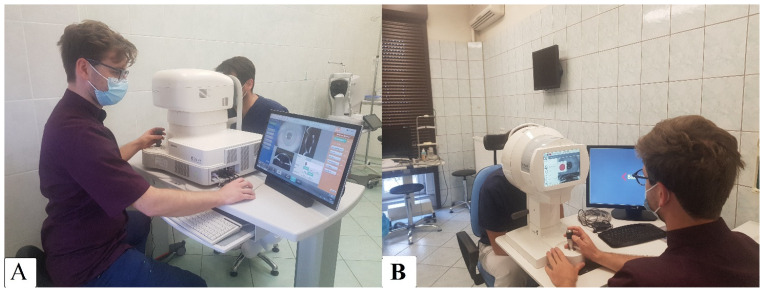
Figure presenting the measurement made with the compared devices. (**A**)—OCT Cassia 2. (**B**)—GalileiG6 Dual Scheimpflug Analyzer.

**Figure 2 diagnostics-13-00267-f002:**
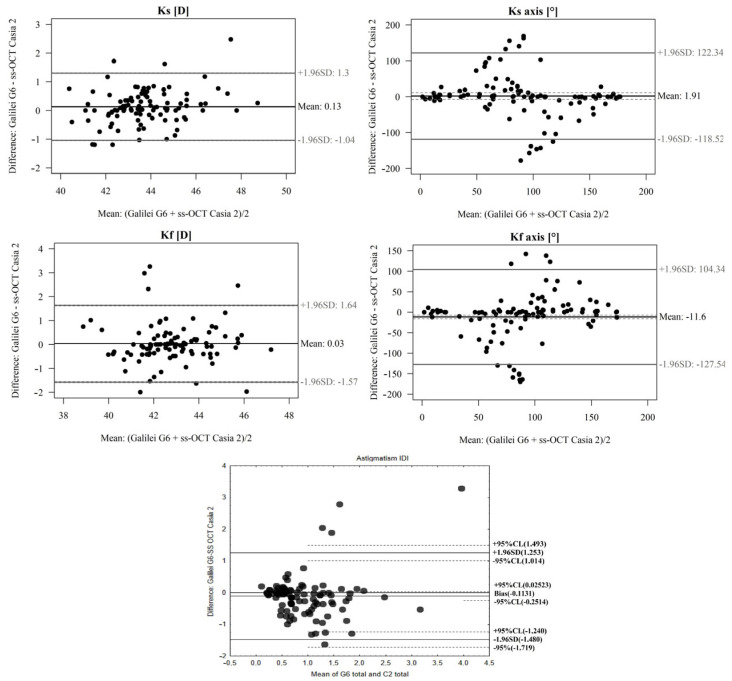
A Bland–Altman plot showing the agreement of total Ks (D), Kf (D), Kf axis (°), Ks axis (°), and total astigmatism As.(D) measurements between the OCT Casia 2 and GalileiG6 Dual Scheimpflug Analyzer. The line shows the mean difference, and the top and bottom dashed lines show the upper and lower 95% LoA, respectively.

**Figure 3 diagnostics-13-00267-f003:**
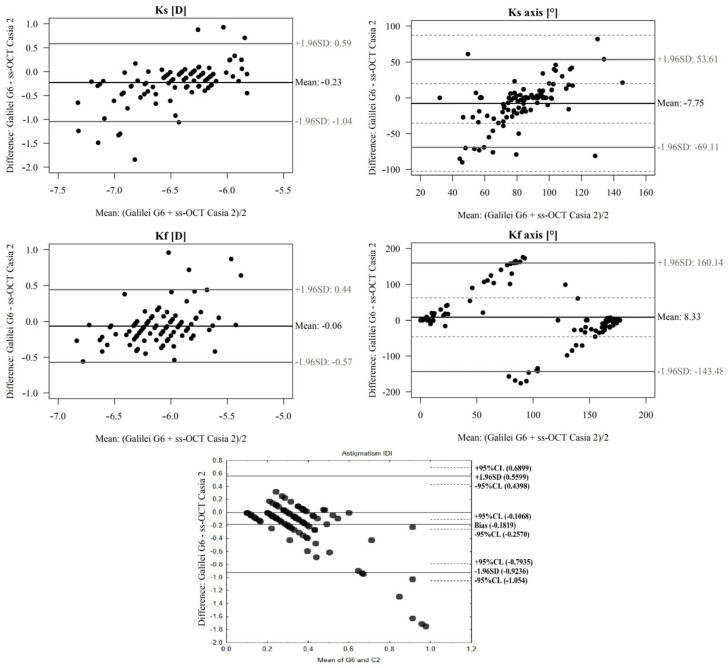
A Bland–Altman plot showing the agreement of posterior Ks (D), Kf (D), Kf axis (°), Ks axis (°), and posterior astigmatism As. (D) measurements between the OCT Casia 2 and GalileiG6 Dual Scheimpflug Analyzer. The line shows the mean difference, and the top and bottom dashed lines show the upper and lower 95% LoA, respectively.

**Figure 4 diagnostics-13-00267-f004:**
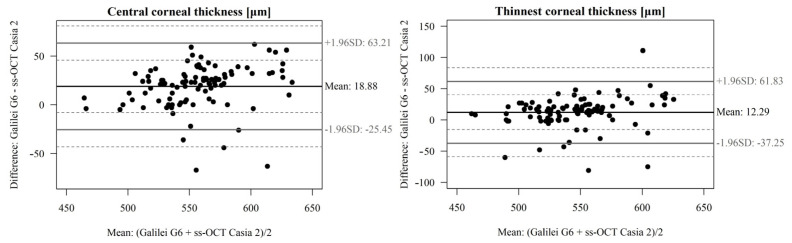
A Bland–Altman plot showing the agreement of central corneal thickness CCT (um) and thinnest corneal thickness TCT (um) measurements between the OCT Casia 2 and GalileiG6 Dual Scheimpflug Analyzer. The line shows the mean difference, and the top and bottom dashed lines show the upper and lower 95% LoA, respectively.

**Table 1 diagnostics-13-00267-t001:** Table comparing total Ks(D) and Kf (D), Ks axis (°), Kf axis (°), total astigmatism As.(D) measured with OCT Casia 2 and GalileiG6 Dual Scheimpflug Analyzer. ICC (introclass correlation coefficient) refers to the correlation between devices.

	GalileiG6	ss-OCT Casia 2	Difference: GalileiG6-ss-OCT Casia 2	*p* *	Limits of Agreement		ICC (95% CI)
					Lower (95% CI)	Upper (95% CI)	
Ks (D) (Mean ± SD)	43.8 ± 1.7	43.67 ± 1.55	0.13 ± 0.6	*p* = 0.083	−1.04 (−1.05; −1.03)	1.3 (1.29; 1.31)	0.93 (0.9; 0.95)
Kf (D) (Mean ± SD)	42.79 ± 1.62	42.76 ± 1.61	0.03 ± 0.82	*p* = 0.643	−1.57 (−1.6; −1.55)	1.64 (1.61; 1.66)	0.87 (0.83; 0.91)
Ks axis (°) (Mean ± SD)	89.47 ± 51.35	87.56 ± 60.8	1.91 ± 61.44	*p* = 0.627	−118.52 (−248.26; 11.23)	122.34 (−7.41; 252.08)	0.41 (0.26; 0.54)
Kf axis (°) (Mean ± SD)	84.07 ± 55.36	95.67 ± 46.45	−11.6 ± 59.15	*p* = 0.108	−127.54 (−247.8; −7.28)	104.34 (−15.92; 224.6)	0.32 (0.17; 0.46)
Ast. (D) (Mean ± SD)	1.00± 0.64	0.89 ± 0.76	0.66± 0.76	*p* = 0.03	−1.48 (−1.72; −1.24)	1.25 (1.49; 1.01)	0.02 (−0.13; 0.17)

* Paired Wilcoxon test.

**Table 2 diagnostics-13-00267-t002:** Table comparing posterior Ks(D) and Kf (D), Ks axis(°), Kf axis(°), posterior astigmatism As.(D) measured with OCT Casia 2 and GalileiG6 Dual Scheimpflug Analyzer. ICC (introclass correlation coefficient) refers to the correlation between devices.

	GalileiG6	ss-OCT Casia 2	Difference: GalileiG6-ss-OCT Casia 2	*p* *	Limits of Agreement		ICC (95% CI)
					Lower (95% CI)	Upper (95% CI)	
Ks (D) (Mean ± SD)	−6.56 ± 0.51	−6.33 ± 0.3	−0.23 ± 0.42	*p* < 0.001	−1.04 (−1.05; −1.04)	0.59 (0.58; 0.59)	0.43 (0.23; 0.59)
Kf (D) (Mean ± SD)	−6.13 ± 0.36	−6.07 ± 0.26	−0.06 ± 0.26	*p* < 0.001	−0.57 (−0.57; −0.57)	0.44 (0.44; 0.44)	0.65 (0.55; 0.74)
Ks axis (°) (Mean ± SD)	79.29 ± 32.08	87.04 ± 17.6	−7.75 ± 31.31	*p* = 0.021	−69.11 (−102.79; −35.43)	53.61 (19.93; 87.29)	0.26 (0.1; 0.4)
Kf axis (°) (Mean ± SD)	104.43 ± 67.06	96.1 ± 80.3	8.33 ± 77.45	*p* = 0.619	−143.48 (−349.66; 62.7)	160.14 (−46.04; 366.32)	0.45 (0.31; 0.57)
Ast. (D) (Mean ± SD)	0.45 ± 0.36	0.27 ± 0.13	0.49 ± 0.36	*p* < 0.001	−0.92 (−0.79; −1.05)	0.56 (0.69; 0.44)	0 (−0.13; 0.14)

* Paired Wilcoxon test.

**Table 3 diagnostics-13-00267-t003:** Table comparing central corneal thickness CCT (um), TCT (um) measured with OCT Casia 2 and GalileiG6 Dual Scheimpflug Analyzer. ICC (introclass correlation coefficient) refers to the correlation between devices.

	GalileiG6	ss-OCT Casia 2	Difference: GalileiG6-ss-OCT Casia 2	*p* *	Limits of Agreement		ICC (95% CI)
					Lower (95% CI)	Upper (95% CI)	
CCT (um) (Mean ± SD)	566.71 ± 39.31	547.83 ± 34.28	18.88 ± 22.62	*p* < 0.001	−25.45 (−43.04; −7.87)	63.21 (45.63; 80.8)	0.72 (0.37; 0.85)
TCT (um) (Mean ± SD)	551.67 ± 39.77	539.38 ± 34.13	12.29 ± 25.27	*p* < 0.001	−37.25 (−59.2; −15.29)	61.83 (39.87; 83.78)	0.73 (0.58; 0.82)

* Paired Wilcoxon test.

## Data Availability

Not applicable.

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
