# Peer review of "Comparative Analysis of Corneal Parameters Performed with GalileiG6 and OCT Casia 2"

_diagnostics, 2023, doi:10.3390/diagnostics13020267_

Round 1
Reviewer 1 Report
This is a helpful scientific evaluation of the accuracy of measurement of two different instruments for preevaluation prior to cataract surgery. The findings should help clinicians with regard to choice of these instruments going forward with regard to accuracy of the measurements. This information will help to avoid post cataract refractive errors.
Author Response
Reply: Thank you for your favorable assessment of our manuscript and the value of our findings with respect to cataract and postoperative refraction. We sent our manuscript for review and correction of English spelling.
Reviewer 2 Report
The experiment design of this study has defects. Therefore, the conclusions from this study are not trustable. From the data showed, in particular, the axis angles of Ks and Kf, the variance are too large. Are these large variations from the patients or from the devices? it's not clear. If the purpose is to compare the measurement differences between two devices, the conditions of the measured objects (here is patients) should be controlled. So the measured variables have relative small variances, like Ks and Kf in this study, this can make sure the result variances are from devices, not from the measured objectives. This reviewer suggested authors to look at the data and group them into subgroups according to variances to make sure the data within the small group has a small variance, then do the comparison between the two measurement devices.
The article's writing needs to be significantly improved for a scientific publication. There are many typos and grammar errors. Some examples are listed below:
1. In the tables, the ICC (introclass correlation coefficient) refers to the correlation within devices or correlation between devices?
2. sample number is inconsistent, there are 68 females and 48 males, total is 116 patients, not 110. But it seems it's 110 patients in total based on the rest of the article.
3. line 122, for ICC explanation, "Excellent" means what? authors should indicate that near 1 means the correlation is excellent. otherwise, how can readers understand "excellent" definition.
4. need explain why some p-values are very low (p<0.05), the correspondent ICCs are also low? for example, table 1, Ks p=0.083, ICC is 0.93, a high value; while in the same table, Ast, p<0.001, while ICC is 0.02, seems conflict. same results happened for other parameters. These definitely need reasons.
5. In table 3, number unit um should be indicated;
6. line 269, sentence grammar mistake.
7. Figure 2, astigmatism x-label is covered.
8. Line 142 to line 144 grammar errors. Don't understand. first say there is significant difference, then say the difference is insignificant. quite confusing.
9. in Study devices characteristic part, in the description of Galilei G6 system, the description is very confusing to this reviewer. If authors can add a figure as a demonstration, it would be very helpful to understand the system.
Overall, this article needs more work before a publication.
Author Response
Reviewer #2: The experiment design of this study has defects. Therefore, the conclusions from this study are not trustable. From the data showed, in particular, the axis angles of Ks and Kf, the variance are too large. Are these large variations from the patients or from the devices? it's not clear. If the purpose is to compare the measurement differences between two devices, the conditions of the measured objects (here is patients) should be controlled. So the measured variables have relative small variances, like Ks and Kf in this study, this can make sure the result variances are from devices, not from the measured objectives. This reviewer suggested authors to look at the data and group them into subgroups according to variances to make sure the data within the small group has a small variance, then do the comparison between the two measurement devices.
The article's writing needs to be significantly improved for a scientific publication. There are many typos and grammar errors. Some examples are listed below:
Overall, this article needs more work before a publication.
Response: Thank you for your comment and the opportunity to correct the error. We carefully checked the number of patients and introduced corrections. 110 patients (62 females and 48 males) participated in the study. We sent the text of the article to the language correction office to correct typos and grammatical errors.
Thanks for the comment and the opportunity to explain. Discrepancies in the results of the Ks Kf axis result from the difficulties in performing measurements by the devices. They are not the result of errors caused by the test object. A similar phenomenon was observed in previous works dealing with similar topics. Looking through the tables with the results, we noticed errors that arose when entering the values into the spreadsheet in the case of the total and rear astigmatism values. We checked the measurements made on the devices. We have entered the correct values in 12 consecutive worksheet panes. We performed the correct statistical calculations of total and posterior astigmatism.
1. In the tables, the ICC (introclass correlation coefficient) refers to the correlation within devices or correlation between devices?
1. Thank you for your comment. ICC refers to the correlation between devices. We added Comments to table descriptions.
2. sample number is inconsistent, there are 68 females and 48 males, total is 116 patients, not 110. But it seems it's 110 patients in total based on the rest of the article.
2. Thank you for your comment and the opportunity to correct the error. We carefully checked the number of patients and introduced corrections. 110 patients (62 females and 48 males) participated in the study.
3. line 122, for ICC explanation, "Excellent" means what? authors should indicate that near 1 means the correlation is excellent. otherwise, how can readers understand "excellent" definition.
3. Thank you for your comment and the opportunity to improve. We added a statement that: The correlation is perfect when its value is close to 1.
4. need explain why some p-values are very low (p<0.05), the correspondent ICCs are also low? for example, table 1, Ks p=0.083, ICC is 0.93, a high value; while in the same table, Ast, p<0.001, while ICC is 0.02, seems conflict. same results happened for other parameters. These definitely need reasons.
4. Thank you for your insightful comment and for pointing out our mistake. We are sorry for the error. We have made corrections. The low ICC was a result of a mistake in the excel spreadsheet, which was la. We have once more checked the data from the original reports, corrected the mistakes, and recalculate ICC.
5. In table 3, number unit um should be indicated;
5. Thank you for your comment. In table 3, we introduced a correction of the um unit.
6. line 269, sentence grammar mistake.
6. Thank you for your important point. We have corrected the grammatical errors in this sentence. We have submitted the manuscript for language editing to remove grammar errors.
7. Figure 2, astigmatism x-label is covered.
7. Thank you for your comment. We corrected Fig. 2. X-label astigmatism is visible.
8. Line 142 to line 144 grammar errors. Don't understand. first say there is significant difference, then say the difference is insignificant. quite confusing.
8. Thank you for your comment. We submitted the manuscript for language editing to remove grammar errors.
9. in Study devices characteristic part, in the description of Galilei G6 system, the description is very confusing to this reviewer. If authors can add a figure as a demonstration, it would be very helpful to understand the system.
9. Thank you for your comment. We have added a picture presenting the measurement made with the compared devices. A - OCT Cassia 2. B - Galilee G6.
Reviewer 3 Report
Patients qualified for cataract surgery were used in measurements one day before scheduled surgical treatment. Total, posterior and corneal power, corneal thickness were assessed by two different ocular devices (Galilei G6 and OCT Casia 2) to perform Bland-Altman analysis. The study aims to provide information about which instrument should be used at different conditions and to help in cataract surgery decision procedure. The results suggest that interchangeable use of the total keratometry parameters can be obtained from both devices. However, parameters, like the axis and total values of astigmatism, central and thinnest corneal measurments and all parameters of the posterior part of the cornea can not be determined interchangeably by two devices used. Explaining the abbreviations like Ks [D] etc would make the manuscript better readable.
Author Response
Reply: Thank you for your favorable assessment of our manuscript and the value of our findings with respect to comparing both devices. We added an explanation of abbreviations, e.g. Ks [D] in place of the first occurrence in the text.. We sent our manuscript for review and correction of English spelling.
Reviewer 4 Report
The authors compare keratometry , astigmatism and its of the posterior surface of the cornea and total, central cornea thickness, and the thinnest corneal 2 thickness measured using two different measurement method. It is interesting , however it requires a discussion on repeatability and riprodusibility and limitations of these machine and comment ability to detect change on published literature (Improving precision for detecting change in the shape of the cornea in patients with keratoconus.)
Author Response
Thank you for your comment and the opportunity to make changes. We added a paragraph about repeatability and riprodusibility to the discussion. We have referred to the detection of corneal disorders such as keratoconus in the discussion.
Round 2
Reviewer 2 Report
This reviewer insisted that repeatability measurements for Ks and Kf axis data are necessary.
Author Response
Thank you for your comment and the opportunity to make corrections. We have included repeatability measurements from the second part of our database for future publication. We made measurements of the Ks axis. We did not include the results of the measurement of the astigatism axis Kf because it is identical. It only has a 90 degree shift.
We added the sentence:
Due to the finding of discrepancies in the measurements of the astigmatism axes Ks and Kf, we performed repeatability measurements of the astigmatism axes for Galilei G6 Dual Scheimpflug Analyzer and CASIA2 Swept Source OCT. The repeatability of the total astigmatism axis Ks for the Galilei G6 Dual Scheimpflug Analyzer was: Mean 86.09, Std.Dev. 86.09, ICC 0.92. The repeatability of the total astigmatism axis Ks for CASIA2 Swept Source OCT was: Mean 86.27, Std.Dev. 57.09, ICC 0.92. The repeatability of the Ks posterior astigmatism axis for the Galilei G6 Dual Scheimpflug Analyzer was: Mean 82.63, Std.Dev. 32.80, ICC 0.92. The repeatability of the posterior astigmatism axis Ks for CASIA2 Swept Source OCT was: Mean 87.54, Std.Dev. 18.46, ICC 0.87.
Round 3
Reviewer 2 Report
I think the paper is fine for a publication.